# Proteomic Analysis Reveals Salicylic Acid as a Pivotal Signal Molecule in Rice Response to Blast Disease Infection

**DOI:** 10.3390/plants11131702

**Published:** 2022-06-27

**Authors:** Haiying Zhou, Delight Hwarari, Yunhui Zhang, Xiaosong Mo, Yuming Luo, Hongyu Ma

**Affiliations:** 1Jiangsu Key Laboratory for Eco-Agricultural Biotechnology around Hongze Lake, Jiangsu Collaborative Innovation Center of Regional Modern Agriculture and Environmental Protection, Huaiyin Normal University, Huai’an 223300, China; hyzhou2018@163.com; 2College of Biology and the Environment, Nanjing Forestry University, Nanjing 210037, China; ton-dehwarr@njfu.edu.cn; 3College of Plant Protection, Nanjing Agricultural University, Nanjing 210095, China; zhyhbio@163.com; 4Jiangsu Grain and Oil Quality Monitoring Center, Nanjing 210031, China; 18900668152@189.cn

**Keywords:** rice, salicylic acid, blast disease, proteomics

## Abstract

Rice blast disease caused by a fungus, *Magnaporthe grisea*, is one of the most destructive diseases in rice production worldwide, and salicylic acid (SA) can efficiently decrease the damage of *M. grisea*. Here, we combined the 2-Dimensional-Liquid Chromatography and the Matrix-assisted laser desorption/ionization time-of-flight mass spectrometer (2D-LC-MALDI-TOF-TOF MS) techniques to compare and identify differentially expressed labelled proteins by the isobaric tags for relative and absolute quantitation (iTRAQ) between the blast-resistant cultivar Minghui and the susceptible rice cultivar Nipponbare in response to blast fungus infection. The group samples were treated with salicylic acid and compared to control samples. A total of 139 DEPs from the two cultivars showed either more than a two-fold change or alternating regulation patterns. Protein functionality analysis also exhibited that these proteins are involved in a wide range of molecular functions including: energy-related activity (30%), signal transduction (11%), redox homeostasis (15%), amino acid and nitrogen metabolism (4%), carbohydrate metabolism (5%), protein folding and assembly (10%), protein hydrolysis (9%), protein synthesis (12%), and other unknown functions (4%). Specifically, we demonstrated that exogenous treatment with salicylic acid promoted recovery in both rice cultivars from *Magnaporthe grisea* infection by enhancing: the regulation of signal transduction, increasing energy conversion and production through the regulation of the glycolytic pathway, and other various biochemical processes. These findings may facilitate future studies of the molecular mechanisms of rice blast resistance.

## 1. Introduction

Rice is a staple crop of economic importance in many countries, as well as being a model monocot species for plant research [1]. With an estimated production capacity of at least 600 million tons per annum, its production is affected by both biotic and abiotic factors. Infectious diseases are among the sternest threats to rice crop production, reducing rice productivity by 25% worldwide [2]. The *Magnaporthe grisea* (Hebert) Barr (anamorph, *Pyricularia grisea* Sacc.), an ascomycete filamentous fungus, one of the most infamous causative agents of blast disease, continuously threatens rice production globally [3], affecting the Poaceae family including rice [4]. Visible disease symptoms are characterized by white to gray-green lesions or spots with darker borders. However, with the enhancement of agronomic practices, several cultural strategies such as the use of genetically resistant varieties and the application of fungicides have been implemented to control blast disease in rice [5,6]. Recently, Zhang et al. [7] reported on a tripartite interaction among rice plants, insects, and pathogens in the control of rice blast disease. They have demonstrated that pre-infestation by herbivores on rice crops reduces the disease incidence of rice blast by increasing the accumulation of salicylic acid (SA) [7], emphasizing a known fact that phytohormones accumulation including SA improves basal crop resistance to blast disease in rice through reducing plant biochemical processes such as: plant photosynthesis rate and regulating the antioxidant defense activities [8,9]. 

Measuring rice proteomic changes in response to salicylic acid (SA) has provided new insights into the molecular mechanisms of SA-induced rice defense to blast disease. It has been shown that SA treatment after rice blast infection elevates rice blast disease resistance. Sun et al. [10] identified 33 protein spots that were differentially expressed after SA treatment, evidencing that SA improves basal resistance to rice blast through decreasing plant photosynthesis, regulating the antioxidant defense activities. They concluded that enhanced blast disease resistance in rice plants is coordinated by SA-induced complex cellular activities. Although, full comprehension of the mechanisms of regulation is still yet to be attained. Cell-level research has also demonstrated that SA signaling is mediated by a series of downstream factors such as *OsNPR1* and *WRK45* acting in parallel, and OsAAA-ATPase1 [11,12,13]. 

The integration of various powerful fields such as proteomics has also enabled the investigation of rice blast disease at the translational level, disclosing several key genes and proteins associated with rice blast resistance [14]. Hence, utilizing proteomic techniques such as two-dimensional liquid chromatography mass combined with the matrix-assisted laser ionization time-of-flight mass spectrometry (MALDI-TOF/TOF MS) for both qualitative and quantitative characterizations of protein mixtures and post-translational modifications has extended the breadth and depth of proteome analysis [15,16,17]. The 2D-LC proteome analysis has been widely used in the identification of protein expression patterns, to address the mechanisms by which *Magnaporthe oryzae* regulates G-proteins (GTP-binding protein) and signaling proteins (RGS) that orchestrate blast disease incidence [18]. Likewise, several researchers have also used the MALDI-TOF/TOF MS in the identification and quantification of protein expression patterns in SA-added rice plants resisting the blast disease.

First developed in 2004, the Isobaric Tags for Relative and Absolute Quantification (iTRAQ) has also been shown to pose unique advantages over other conventional proteomics techniques [19]. It enables the identification and quantification of many proteins from specific biological environments using labeled peptides as internal standards identifiable by sensitive mass spectrometers. Furthermore, it utilizes bioinformatics tools and statistical packages to visualize obtained data. The incidence of the iTRAQ approach has enabled many researchers to identify proteins involved in pathogen resistance in several plant species. Ma et al. [20] implemented the iTRAQ to compare and detect differentially expressed proteins (DEPs) from a blast-resistant variety and a susceptible variety. They detected 46 common DEPs among the two varieties involved in plant-pathogen interaction, plant hormone signal transduction, fatty-acid metabolism, and peroxisome biosynthesis [20]. Furthermore, iTRAQ has been implemented in several kinds of research that may include the quantification of rice grains proteins [19], assessing the effect of high night temperature stress on rice [21], assessing the impact of ion pollution and detoxification in rice [22], and assessing the expression patterns of rice proteins under several abiotic stresses [23,24]. 

Although the iTRAQ approach has been utilized in the past to assess proteins related to rice blast resistance, few proteins have been successfully identified. In this present study, we compared the posttranslational responses of rice cultivars (blast-resistant Minghui and blast-susceptible Nipponbare cultivars) sprayed with salicylic acid against the controls to *Magnaporthe grisea* fungus infection using the iTRAQ approach to label the peptides and the combined 2D-LC/MALDI-TOF-TOF MS technique for protein separation and identification, with an intent to comprehend the molecular mechanisms underlying blast resistance in rice. This study may facilitate and provide valuable molecular data for rice blast-resistant breeding purposes.

## 2. Results

### 2.1. Identification and Relative Quantification of Proteins

In present study, differentially expressed proteins (DEPs) expressed in response to *Magnaporthe grisea* (M) infection were identified using the isobaric Tags for Relative and Absolute Quantitation (iTRAQ) technique coupled with the 2-D Liquid Chromatography-Matrix-assisted laser desorption/ionization time-of-flight mass spectrometer (2D-LC-MALDI-TOF-TOF MS) analysis. The proteomic differences were contrasted and analyzed from the two rice cultivars, the blast-resistant Minghui (M) and blast susceptible Nipponbare (R) varieties, samples were grouped in four sub-pairs as M-M vs. M-CK, M-SA vs. M-CK, R-M vs. R-CK, and R-SA vs. R-CK. The schematic workflow of the experimental design is shown in Figure 1. A total of 1230 proteins were obtained, which were further separated using the Liquid Chromatography (LC), and 1163 proteins were successfully identified using the MASCOT program. Expression analysis of the observed proteins exhibited that 139 proteins had more than 2.0-fold change and/or less than 0.5-fold difference with *p* < 0.05 between samples, and these were considered to be significantly altered, as shown in Table 1.

These DEPs were shown to be involved in a wide range of molecular functions, 30%, which was the majority, was constituted in energy-related activities, 11% in signal transduction, 15% in redox homeostasis, 4% in amino acid and nitrogen metabolism, 5% in carbohydrate metabolism, 10% in protein folding and assembly, 9% in protein hydrolysis, 12% in protein synthesis, and the remaining 4% in other unknown functions (Figure 2). This suggests that biological processes with high protein constituency such as energy-related activities are likely to enhance blast disease resistance in rice varieties. A deeper analysis in the energy-related activity showed that 65 of the total DEGs were upregulated, to which 36 and 29 were enriched in Minghui and Nipponbare, respectively. Specifically, the 23 kDa polypeptide of photosystem II, chloroplastic glutamine synthetase, and enoyl-CoA hydratase/isomerase proteins were highly upregulated among the significantly expressed protein in the R-SA group and downregulated in the R-M group in the Nipponbare variety. 

### 2.2. Bioinformatics Analysis of Differentially Expressed Proteins

To define the key proteins associated with rice blast resistance and SA-dependent recovery, we did bioinformatics analyses of the four pair samples, M-M vs. M-CK, M-SA vs. M-CK, R-M vs. R-CK, and R-SA vs. R-CK. The differentially expressed analysis of most significant protein changes between the two varieties showed that three common DEPs were expressed in all the groups (Figure 3). This implies that these DEPs may play an important role in all rice cultivars of study and also that there is less in common in the regulation of blast disease infection between the two varieties of study. As expected, the SA-added groups, M-SA and R-SA, made up the largest part of the differentially expressed proteins, 24 and 20, respectively, compared to the control groups. This shows that salicylic acid had an important effect on fungal infection rates. On the other hand, the R-M group also constituted a larger group of DEPs (24) as compared to the M-M group, suggesting that the R- group (Nipponbare variety) was negatively affected by blast disease. Comparisons between two cultivars showed that more DEPs were expressed in the susceptible Nipponbare cultivar as compared to the blast-resistant Minghui cultivar, evidencing the fact SA exhibited positive regulatory roles against the *M. grisea* infection. Hence, these results can be summarized as that salicylic acid upregulated the expression of DEPs in both varieties.

### 2.3. Comprehensive Inventory of the Differentially Expressed Proteins 

To classify the DEPs in Minghui and Nipponbare varieties, putative proteins induced by *M. grisea* were separated by the 2Dimensional- Liquid Chromatography (2D-LC) (Figure 4). All the identified DEPs were classified into three GO terms; biological process (BP), molecular function (MF), and cell component (CC). The majority of the identified proteins in the M-M group were involved in single-organism metabolic processes (BP analysis), cytoplasm (CC analysis), and nucleoside phosphate binding (MF analysis). The identified proteins in the M-SA group were concentrated in the organonitrogen compound metabolic process (BP analysis), cytoplasm (CC analysis), oxidoreductase activity, and catalytic activity (MF analysis). In the R-M group, the majority of identified proteins were involved in the small molecule metabolic process (BP analysis), cytoplasm (CC analysis), and catalytic activity (MF analysis). Lastly, the identified proteins in the R-SA group were concentrated in the organonitrogen compound metabolic process (BP analysis), chloroplast and cytoplasm (CC analysis), oxidoreductase activity, and nucleoside phosphate binding (MF analysis). Wholly, these results show that DEPs from different rice cultivars and exposed to different treatments exhibited different functional characteristics, and that more genes were assigned in the susceptible Nipponbare cultivar as compared to the blast-resistant Minghui cultivar.

### 2.4. Specific Metabolic Processes in Minghui and Nipponbare Are Involved in the Response against M. grisea

To determine specific metabolic processes in Minghui and Nipponbare following *M. grisea* inoculation and salicylic acid spray, we used the KEGG database (Figure 5). We noted that only three metabolic pathways were significant at both *p*-values (*p* < 0.01 and *p* < 0.05), including carbon fixation and metabolism, carbon metabolism, glyoxylate and dicarboxylate metabolism. Carbon fixation and metabolism exhibited the highest −log(Pvalue) change in both groups of the Minghui cultivar, while the carbon metabolism, glyoxylate and dicarboxylate metabolism were highly upregulated in the R-M and R-SA groups, respectively. However, significant expression differences were observed when comparing two distinct varieties with the same treatment. In SA-treated cultivars we noted that 50% of the biochemical processes in the M-SA group were not significant at *p*-value < 0.05 as compared to the R-SA group, to which only two of the investigated processes were insignificant when *p*-value < 0.05. In the *M. grisea*-treated groups, similar findings were observed, although all the processes investigated in the R-M group had significant differences when *p*-value < 0.05, except for the Glycine, serine and threonine metabolism in the M-M group. Based on the results of the KEGG analysis, the salicylic acid added groups from both varieties had higher upregulation expression as compared to the control groups. 

### 2.5. Protein-Protein Interactions

To comprehend the protein network interaction, we constructed protein to protein interactions (PPI) using the STRING database and viewed using Cytoscape software (Figure 6). Generally, the results exhibited that the identified proteins were densely connected and regulated several pathways. In the M-M (Figure 6A) group, the majority of the investigated proteins with high fold change were involved in metabolic pathways, moderately upregulating its enrichment. The carbon fixation in photosynthetic organisms and carbon metabolism pathways were highly enriched with fewer moderately upregulated proteins. The glyoxylate and dicarboxylate metabolism was lowly enriched by several upregulated proteins. Likewise, the glycine, serine, and the threonine metabolism were upregulated and downregulated by two proteins, 2,3-diphosphoglycerate-independent and betaine aldehyde dehydrogenase, respectively. Similar enrichment patterns were observed in the M-SA group (Figure 6B), with fewer proteins involved in the metabolic pathways, and highly upregulated as compared to the M-M group. Specifically, the metabolic pathways were densely connected with several upregulated proteins. However, some of the pathways were absent in the M-M group and were regulated by several proteins. For instance, the pentose phosphate pathway was present in M-SA but absent in R-SA and was lowly enriched by two upregulated proteins. Intriguingly, vitamin B6 metabolism was not linked to any of the pathways within the network, and only one protein was highly upregulating its enrichment. 

The R-M group (Figure 6C) was also densely connected with moderately expressed proteins. Carbon metabolism was highly enriched among the observed pathways, with fewer proteins involved as compared to the metabolic pathways. Nonetheless, the metabolic pathways were moderately enriched with two highly expressed proteins involved. The remaining pathways in the R-M group, oxidative phosphorylation, pyruvate metabolism, and arginine biosynthesis, were lowly enriched and had two, three, and four downregulated proteins, respectively, linking their enrichment. The R-SA group was sparsely connected (Figure 6D), with some proteins highly expressed while some were lowly expressed. The majority of the highly expressed proteins with a high fold change moderately enriched the regulation of metabolic pathways, with a few downregulating their enrichment. Nonetheless, the carbon fixation in photosynthetic organisms and glyoxylate and dicarboxylate metabolisms were highly enriched with two and three proteins, respectively, linked.

## 3. Discussion

Developments in proteomics research over the past decades have enabled the comprehension of plant disease resistance at the protein level. In this study, the proteomics approach was employed to comparatively identify the expressed proteins from the blast-susceptible Nipponbare and the blast-resistant Minghui rice cultivars. To reveal the mechanism of resistance to *Magnaporthe grisea* infection, the two rice cultivars were treated with the biotic stress hormone, salicylic acid, after infection. A total of 139 DEPs were identified among all samples and were further shown to be involved in energy metabolism, signal transduction, carbohydrate metabolism, nitrogen metabolism, protein hydrolysis, and the synthesis process. 

### 3.1. Differentially Expressed Proteins Related to Energy

Energy deficits are a general symptom of photosynthetic plants under stress due to reductions in photosynthesis and/or respiration, ultimately resulting in cell death and growth arrest. However, plants counter energy shortages by enhancing inherent pathways of carbohydrate metabolism, induction of alternative pathways of glycolysis, and maintaining energy supply and carbon skeletons for key metabolic processes. Sucrose is the primary translocated carbohydrate in the majority of plants; thus, its metabolism is vital in the regulation of plant growth in response to stresses [25,26,27]. We identified 41 energy-related proteins (spots 1–41) involved in the response to blast fungus infection in rice. These proteins mainly included 23 kDa polypeptide of photosystem II, chloroplastic glutamine synthetase, NADP-dependent malic enzyme, oxalate oxidase-like protein, uroporphyrinogen decarboxylase, 6-phosphogluconolactonase, 2,3-bisphosphoglycerate-independent phosphoglycerate mutase, and ATP synthase (Table 1). In plants, these proteins are involved in the glycolysis pathway, tricarboxylic acid cycle, chlorophyll synthesis, pentose phosphate pathway, photosynthesis process energy, and so on. 

ATP synthase is widely distributed in the mitochondrial membrane and chloroplasts thylakoid [28], participating in oxidative phosphorylation and photophosphorylation, under the impetus of the transmembrane proton power potential, resulting in ATP hydrolysis and the release of energy, which is the most direct energy source in plants [29,30]. We observed that the levels of ATP synthase proteins significantly declined under the blast fungus infection in Nipponbare (susceptible cultivar) and that treatment with salicylic acid failed to promote significant recovery of ATP synthase protein levels in Nipponbare. Nonetheless, in Minghui (resistant variety) the three levels of ATP synthase proteins spots (13, 14, and 16) were slightly downregulated under blast fungus infection, and treatment with salicylic acid significantly elevated the levels of these proteins. These results demonstrated that *Magnaporthe grisea* infection significantly reduces energy production in Nipponbare compared to Minghui and that salicylic acid exogenous treatment promotes more energy recovery in Minghui than in Nipponbare after infection. 

Thirteen differentially expressed protein spots (1, 5, 24, 28, 30–36, 39, 40) were involved in photosynthesis metabolism, including chlorophyll synthesis; in Minghui under blast fungus infection these proteins were upregulated but not in Nipponbare. Treatment with salicylic acid afterwards further improved the expression of these proteins in Minghui. We identified five proteins (spots 3, 20, 22, 25, 26) involved in the glycolytic pathway. Glycolysis is the process of degrading glucose to produce ATP, providing partial energy for cell activities. In this study, the expression of proteins involved in the glycolytic pathway was downregulated in Nipponbare infected by *Magnaporthe grisea*. However, in Minghui two of the five proteins (spots 22 and 26) were upregulated significantly. These results showed that the glycolytic pathway was inhibited by blast infection in Nipponbare. 

### 3.2. Proteins Associated with Signal Transduction

Stress signal perception and transmission are important for plant environmental stress response [31]. We identified 15 signal transduction-related proteins, including brain-specific protein, calreticulin precursor, and GTP-binding protein (Table 1). In Nipponbare, most of the expressed proteins were downregulated, while in Minghui the expressed proteins were upregulated. This implies that active regulation of signal transduction is associated with blast infection resistance in Minghui. 

Integrins belong to a family of cell surface glycoproteins, which connect the extracellular matrix to the actin cytoskeleton and serve in cell signaling across the plasma membrane to confer cell survival. Overexpressed 14-3-3 beta protein interacts with cytosol B1 integrin promoting cell migration and signal transmission [32,33]. We analyzed the protein level of 14-3-3 beta (spot 50) and noted that its stability was constant in Minghui, while it was downregulated in Nipponbare under blast fungus infection. Spot 59 is a spermidine synthase important in plant stress responses. It is involved in signal transduction associated with plant stress resistance [34]. The expression of the protein was significantly upregulated only in Nipponbare after treatment with salicylic acid. Wholly, this result showed that SA enhances the regulation of signal transduction by up- and downregulating proteins involved. However, the correlation between proteomic changes due to rice blast resistance may deserve further investigation.

### 3.3. Differentially Expressed Proteins with Redox Homeostasis 

There is active redox homeostasis in a plant cell in response to stress stimuli. Oxidants mainly include reactive oxygen and nitrogen species, and reductants (antioxidant molecules). It has been shown that many enzymes are involved in redox homeostasis. Normal cell functions require the balance of oxidation and reduction processes [35,36]. We identified 21 proteins (spots 57–77) associated with redox homeostasis under blast fungus infection, which mainly included glutathione S-transferase, putative glyoxalase I, manganese superoxide dismutase, lactoylglutathione lyase, and superoxide dismutase (Table 1). In general, most of these proteins in Nipponbare were downregulated under the blast fungus infection. In Minghui, 5 out of the 21 proteins were upregulated and 3 downregulated under the blast infection. Treatment with salicylic acid did not change their expression patterns, suggesting the presence of an SA-mediated defense in the resistant cultivar subjected to blast infection. 

Glutathione S-transferase is the key enzyme of glutathione binding reaction, and catalyzes the initial steps of the glutathione-binding reaction. It can catalyze the binding reaction of nucleophilic glutathione with various electrophilic exogenous chemicals. Many exogenous chemicals are easily separated from some bioactive intermediates in the biotransformation of the first phase reaction; they can covalently combine with the important components of cell biological macromolecules and cause cell damage. Glutathione can prevent the occurrence of such covalent binding for detoxification [37,38]. In this study, the expression of these proteins in both varieties decreased, but the Minghui cultivars showed a recovery after treatment with salicylic acid (spot 43). This suggests that exogenous treatment with salicylic acid is necessary to induce the glutathione-binding reaction process in Minghui after infection.

Manganese superoxide dismutase (spot 60) is an important antioxidation enzyme [39], alongside Betaine aldehyde dehydrogenase (spot 66) involved in the synthesis of betaine, which is the substance in methylation reactions and detoxification of homocysteine [40]. PDI-like protein (Spot 74) is also crucial; it catalyzes the formation of disulfides and also regulates specific biochemical processes [41]. The expression of these three proteins was significantly upregulated in Minghui infected with *Magnaporthe grisea*. By contrast, in Nipponbare, expressions of the proteins were significantly downregulated under blast fungus infection. 

### 3.4. Proteins Associated with Amino Acid and Nitrogen Metabolism 

Previously, rice blast fungus infection was shown to alter expression patterns of proteins involved in amino acid metabolism, nitrogen assimilation, nucleotide anabolism, and lipid oxidation [42]. In this study, we observed that cysteine synthase, glutamate ammonia ligase, ketol-acid reductoisomerase, and ornithine acetyltransferase expressions were altered in response to blast fungus infection (Table 1). The activities of cysteine synthase determine the level of cysteines, an important amino acid in protein synthesis. The enzyme plays a very important role in antioxidation and stress resistance in plants [43,44]. In Nipponbare, Cysteine synthase was downregulated after blast fungus infection, and recovery was observed after treatment with salicylic acid. In Minghui, the expression of cysteine synthase remained stable under fungus infection.

Glutamate ammonia ligase (spot 81) plays a role in ammonia and glutamate detoxification, acid-base homeostasis, cell signaling, and cell proliferation. Ketol-acid reductoisomerase (spot 82) is one of the key enzymes in the biosynthesis of amino acids with branched chains in plants; however, interruption of the synthesis can cause plant death [45]. Ornithine acetyltransferase (spot 83) regulates the abundance of L-arginine, which plays an important role in maintaining cell functions [46]. Generally, the expressions of these three proteins were inhibited in Nipponbare under the *Magnaporthe grisea* infection; additional treatment with SA improved the expression of these proteins in blast-infected Nipponbare, while in Minghui their levels remained stable under infection.

### 3.5. Proteins Associated with Carbohydrate Metabolism

Glucose, a monosaccharide metabolized in plants, is one of the most important carbohydrates in plant cell metabolism [47]. We observed that the expression patterns of seven proteins involved in carbohydrate metabolism were downregulated in response to blast fungus infection in Nipponbare. Additional treatment with SA elevated the expression of these proteins, evidencing the vital role of SA during blast fungus infection. Furthermore, the cytoplasmic malate dehydrogenase (spot 86) plays an important role in the shuttle system of the cytoplasm and organelles, nucleic acid selective channels, gluconeogenesis, and malic acid metabolism [48]. In Nipponbare, this enzyme was significantly downregulated after *Magnaporthe grisea* infection, suggesting inhibition of corresponding biological processes. In comparison, enzyme levels remained unchanged in Minghui. 

S-adenosylmethionine synthetase 2 (spot 90) plays an important role in cell metabolism in all organisms. We observed that the level of S-adenosylmethionine synthetase was downregulated only in Nipponbare infected by *Magnaporthe grisea*, and later recovered with salicylic acid treatment. Takusagawa et al. [49] have shown the abundance of spot 90 in more than one hundred different kinds of methyltransferase catalytic reaction of a donor in the synthesis of glutathione (GSH) precursor molecules of the sulfur transfer process and polyamine synthesis to aminopropyl process, and also related activity to a variety of enzymes [49].

### 3.6. Differentially Expressed Proteins Related to Protein Metabolism

Thirty-three proteins (spots 91–133) were shown to be mainly involved in the processes of protein metabolism including protein synthesis, protein folding, and assembly. Proteins involved in protein folding and assembly mainly included heat shock proteins (HSPs), HSP 70 kDa and 90 kDa. Heat shock proteins (HSPs) are synthesized under various stresses [50]. These proteins mainly serve as molecular chaperones to stabilize target proteins subjected to stresses, thereby playing a protective role [51]. In Nipponbare, these proteins were significantly downregulated after infection with *Magnaporthe grisea*. However, in Minghui the expression of these proteins was upregulated. Proteins involved in protein hydrolysis mainly include aminopeptidase, proteasome, hydrolase, and so on. These enzymes mainly function to degrade target proteins. In Nipponbare, these proteins showed a downward regulation trend in general, whereas the opposite was true in Minghui: of the two species, only a few were exceptions. Similar to the proteins involved in protein folding and assembly discussed above, the levels of the proteins may also indicate the differential response of the two rice cultivars under infection. 

Proteins involved in protein synthesis mainly include ribosomal protein (RP), binding protein, assembly protein, and so on. Ribosomal protein is ubiquitous in every cell and forms ribosomes with RNA to function in protein synthesis [52]. In Nipponbare, the proteins were downregulated in response to blast fungus infection, but an afterward treatment with salicylic acid significantly improved the levels of the proteins. By contrast, in Minghui, the proteins were upregulated after infection. 

## 4. Materials and Methods

### 4.1. Plant Materials

Two rice cultivars, blast-resistant Minghui (M) and blast-susceptible Nipponbare (R), were comparatively assessed for differential expression of proteins (DEPs) in this study. The 20 seeds disinfected with 70% alcohol and rinsed with distilled water were planted into a 10 cm-diameter plant pot in a greenhouse under optimum conditions and supplemented with the nutrient solution. The experiment was designed in two blocks (2 cultivars), 3 treatments (including the control-CK) and 3 replicates. Nine two-week-old rice plants from Minghui were randomly selected and divided into three replicate groups designated as the Minghui set as a control (M-CK), Minghui infected with *Magnaporthe grisea* fungus (M-M), and Minghui sprayed with salicylic acid (M-SA). Likewise, 3 Nipponbare replicate groups were randomly selected and designated as the Nipponbare set as a control (R-CK), Nipponbare infected with *Magnaporthe grisea* fungus (R-M), and Nipponbare sprayed with salicylic acid (R-SA). The groups, R-M, R-SA, M-M, and M-SA, were sprayed with a fungus strain of *M. grisea*, Guy11, at an intensity of 5 × 10^4^/mL. The groups R-CK and M-CK were sprayed with water and served as controls. To investigate the effect of salicylic acid, the groups R-SA and M-SA were sprayed with salicylic acid with a concentration of 8 mmol/L. Then, the rice plants were left for another week to grow. One-week-old treated leaves were harvested and stored at −80 °C before further experiments. 

### 4.2. Protein Extraction

The trichloroacetic acid (TCA)/acetone precipitation method was used to extract proteins [18]. Leaf tissues were ground into a fine powder in liquid nitrogen. The resulting leaf powder was suspended in 10% (*w*/*v*) TCA/acetone supplemented with 0.1% DTT, and stored at −20 °C for 2 h for promoting perception. The suspension was centrifuged at 15,000 rpm for 20 min. The resulting pellet was re-suspended in 10% (*w*/*v*) TCA/acetone supplemented with 1 mM PMSF and 0.07% (*w*/*v*) β-mercaptoethanol, and was stored at −20 °C for 1 h. The suspension was further centrifuged at 15,000 rpm for 20 min to precipitate proteins, which were then subjected to vacuum freeze-drying processes. The resulting protein powder was dissolved in 800 μL lysis solution containing 7 M urea, 2 M thiourea, 4% (*w*/*v*) CHAPS, 65 mM DTT, 1 mM PMSF, and 0.5% (*v/v*) violates. The protein concentration of the sample was quantified according to the Bradford method [53]. 

### 4.3. Trypsin Digestion, iTRAQ™ Labels, Off-Line Strong Cation Exchange Chromatography, and Online nano-LC-MALDI-TOF-TOF MS Analysis

A total of 100 μg of proteins from each sample were subjected to reduction, alkylation, and subsequent tryptic digestion at 37 °C overnight. Afterward, the samples M-CK, M-M, M-SA, R-CK, R-M, and R-SA were labeled with iTRAQ reagents (AB Sciex, Framingham, MA, USA), named 113, 114, 115, 116, 117, and 118, respectively. The 6 samples were then desalted and dried (Figure 1) [54,55]. The iTRAQ-labeled peptides were separated by conducting strong cation exchange (SCX) chromatography on a polysulfoethyl column (Poly-SEA 5μ 300 Å 2.0 × 150 mm, Michrom, Auburn, CA, USA). A total of 10 SCX components were collected and concentrated [56,57]. The samples were subsequently loaded onto the Eksigent nanoLC-Ultra™ 2D chromatographic (Sciex, USA) and Proteineer fc II (Bruker Daltonics Inc., Billerica, MA, USA) systems. The resulting tryptic peptides were dripped onto the MTP AnchorChipTM 384 BC target plate and mixed with 5 mg/mL CHCA supplemented with 0.1% TFA and 50% acetonitrile. Matrix-assisted laser desorption/ionization time-of-flight mass spectrometer (MALDI-TOF-TOF MS) analyses were conducted using an ultrafleXtreme MALDI-TOF-TOF instrument (Bruker Daltonics Inc., USA). The mass range 700–3500 Da reflects positive PMF, and 40–1015 Da for lift [58].

### 4.4. Database Search and iTRAQ Quantification

The spectra of proteins were used for the NCBInr database search with the online MASCOT program (http://www.matrixscience.com, accessed on 2 January 2019). The searching parameters included 0.15 Da mass tolerance for peptides and 0.25 Da mass tolerance for TOF-TOF fragments, one trypsin miscleavage allowed, carbamidomethyl of Cys as fixed modification, and oxidation of Met, pyro-Glu formation of N-terminal Gln and Glu as variable modification. Only significant hits, as defined by the MASCOT probability analysis (*p* < 0.05), were accepted [59]. Quantification of peptides and proteins was based on signature peak areas (*m/z*: 113, 114, 115, 116, 117, and 118). Relative quantification ratios of the identified proteins were calculated, averaged, and corrected for any systematic errors caused by iTRAQ™ labeling of peptides. The accuracy of each protein ratio was given by a calculated error factor in the analysis software. The standard of a 95% uncertainty range was adopted [60]. Finally, a more than 2.0-fold or less than 0.5-fold difference (*p* < 0.05) in expression values was considered as a significant difference in expression.

## 5. Conclusions

The *M. grisea* (Hebert) Barr continuously threatens rice production globally, and exogenous application with salicylic acid enhances rice resistance to blast disease infection. In this study, the iTRAQ approach was utilized to investigate the alteration of global protein expression abundance affected blast infection and SA application in rice varieties with different resistance to blast. A total of 139 DEPs were identified and were mainly involved in a wide range of biological processes, including energy-related activity, signal transduction, redox homeostasis, amino acid and nitrogen metabolism, carbohydrate metabolism, protein folding and assembly, protein hydrolysis, protein synthesis, and others. Comparing their expression patterns in two rice cultivars, susceptible Nipponbare and resistant Minghui, treated with SA and untreated with SA, it was found that expression patterns recovered from disease effects after SA exogenous application, which evidenced the potential role of SA to increase rice blast disease resistance. Nonetheless, there are still gaps within the comprehension of the SA mechanism in increasing blast resistance in rice. By contrast, SA-medicated pathways appeared to be constitutively expressed in the resistant Minghui cultivar, which enhances resistance to blast fungus infections.

## Figures and Tables

**Figure 1 plants-11-01702-f001:**
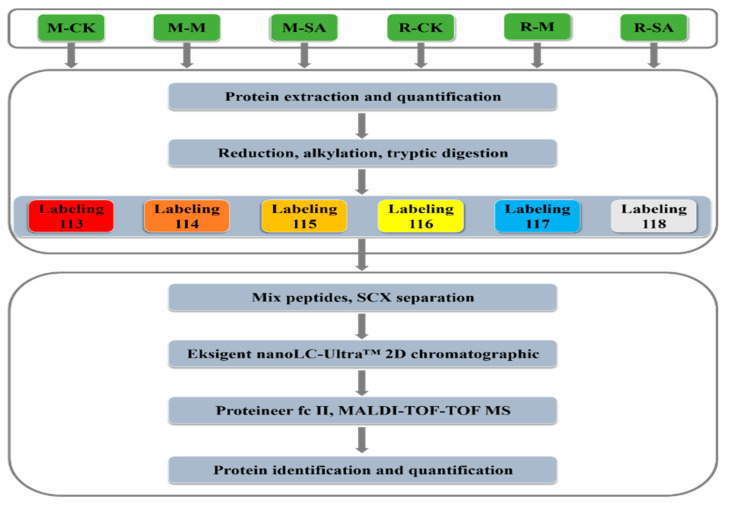
The flowchart depicts the main steps of protein expression analysis in the rice blast-resistant Minghui and the blast susceptible Nipponbare cultivars by conducting six-plex isobaric tagging and the nano Liquid Chromatography-Matrix-assisted laser desorption/ionization time-of-flight mass spectrometer (nanoLC-MALDI-TOF-TOF MS). The treatment groups in the flow chart were abbreviated as M-CK, M-M, and M-SA for the Minghui cultivar sub-groups set as a control, infected with *Magnaporthe grisea*, and sprayed with salicylic acid, respectively. Treatment samples of the susceptible Nipponbare cultivar were also abbreviated as R-CK, R-M, R-SA for samples set as control, infected with *Magnaporthe grisea*, and sprayed with salicylic acid, respectively.

**Figure 2 plants-11-01702-f002:**
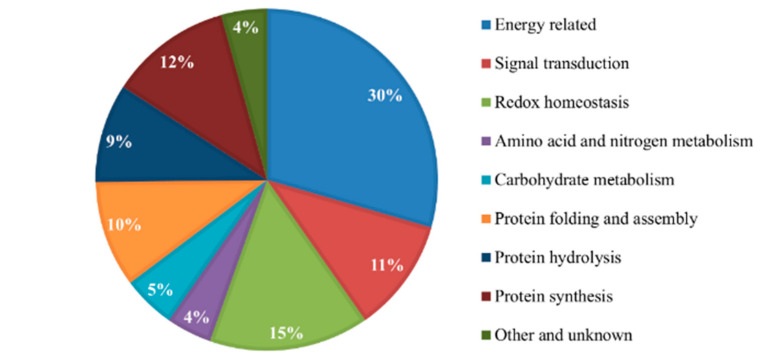
Shows the classification of the 139 proteins expressed differentially in two rice cultivars, blast-resistant Minghui and blast-susceptible Nipponbare.

**Figure 3 plants-11-01702-f003:**
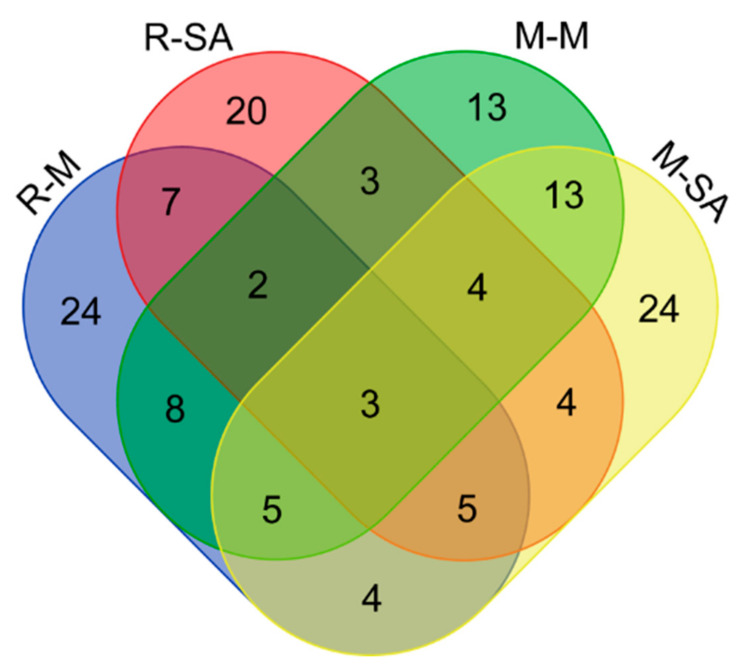
Venn diagram of differentially expressed proteins identified in four experimental groups. Different color schemes denote different group sets, that is, Minghui added with SA (M-SA), Minghui inoculated with *M. grisea* (M-M), Nipponbare added with SA (R-SA), and Nipponbare inoculated with *M. grisea* (R-M).

**Figure 4 plants-11-01702-f004:**
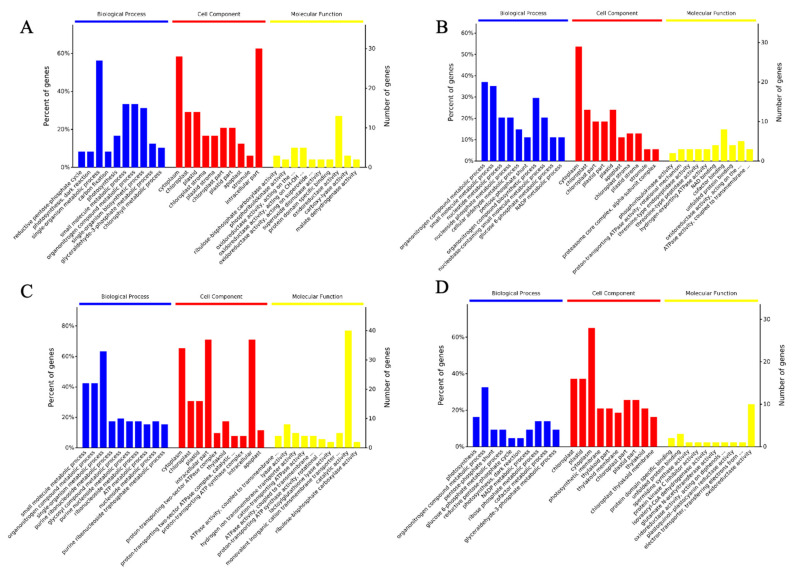
GO annotation of four groups of proteins in three categories: biological process (BP), cellular component (CC), and molecular function (MF). The figures show total percentage of genes assigned to three categories of enrichment analysis of the top ten entries according to the significant degree: biological processes (blue), cell component (red), and molecular function (yellow). From left to right is under the −log (*p*-value) from a large to a small array, which means that the closer to the left side, the more significant. The vertical axis represents the number and proportion of proteins or genes in each entry. (**A**) M-M, (**B**) M-SA, (**C**) R-M, (**D**) R-SA.

**Figure 5 plants-11-01702-f005:**
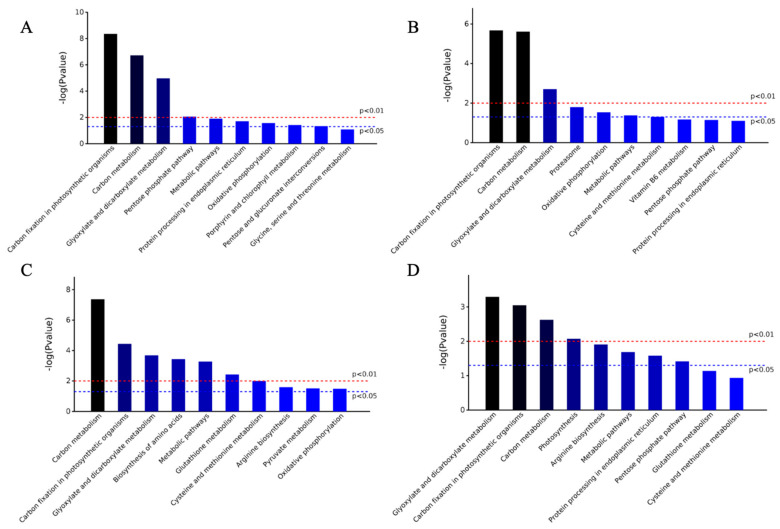
KEGG analysis of four groups of proteins. The figure shows the top ten entries according to a significant degree. From left to right was the following −log(*p*-value) from a large to a small array, which meant that the closer to the left side, the more significant. The vertical axis represents −log (*p*-value). The red line indicates that *p* = 0.01, while the blue line indicates that *p* = 0.05. Only when the column is higher than the red line or blue line, does its corresponding metabolic pathway have significant meaning. (**A**) M-M, (**B**) M-SA, (**C**) R-M, (**D**) R-SA.

**Figure 6 plants-11-01702-f006:**
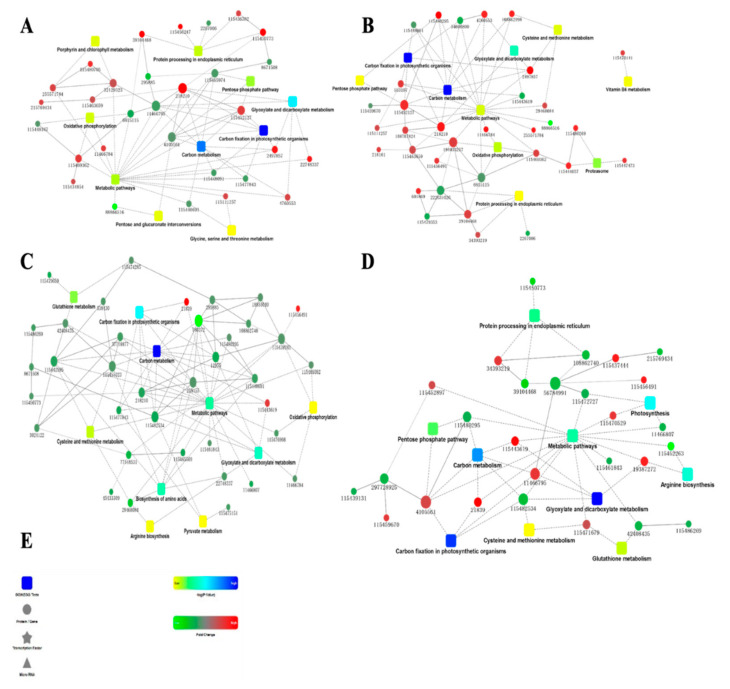
The protein-protein interaction of differentially expressed proteins. The figures show the top ten entries according to the same significant degree as the KEGG analysis (Figure 5). (**A**) Represents the M-M group. (**B**) The M-SA group. (**C**) The R-M group. (**D**) The R-SA group. (**E**) Marginal data of protein-protein interaction results. The dots represent proteins; red means upregulated; green means expression. The rounded rectangles represent the enriched pathways, color from yellow to blue gradient indicates significance from low to high, respectively. A solid line indicates the protein interaction that had been reported, and dotted line that had no experimental results. The ID or names of dots and rounded rectangles are displayed below.

**Table 1 plants-11-01702-t001:** The mascot search results and relative volume (RV) of 139 proteins expressed differentially in two rice cultivars, blast-resistant Minghui and blast susceptible Nipponbare.

^a^ Spot No.	^b^ Accession	^c^ Protein name	^d^ M	^e^ SC	^f^ MP	^g^ Tp*I*	^h^ Tmw	Nipponbare			Minghui		
								^i^ R-CK	^i^ R-M	^i^ R-SA	^i^ M-CK	^i^ M-M	^i^ M-SA
**Energy related**													
1	gi|115470529	23 kDa polypeptide of photosystem II, rice	237	19%	4	8.66	27.1	1.0	0.96	2.82 *	1.0	1.42	1.84
2	gi|19387272	Chloroplastic glutamine synthetase, rice	405	14%	6	6.18	49.8	1.0	1.02	3.68 *	1.0	1.19	1.28
3	gi|54606800	NADP dependent malic enzyme, rice	274	9%	5	5.79	65.8	1.0	0.52	1.26	1.0	0.97	0.19 *
4	gi|115475413	Oxalate oxidase-like protein, rice	227	13%	2	5.48	24.7	1.0	0.88	0.48 *	1.0	0.83	2.13 *
5	gi|115452897	Uroporphyrinogen decarboxylase, rice	96	5%	3	6.15	43.0	1.0	0.73	2.19 *	1.0	1.36	0.96
6	gi|115440691	2,3-bisphosphoglycerate-independent phosphoglycerate mutase, rice	645	17%	8	5.42	61.0	1.0	0.34 *	1.34	1.0	1.66	1.35
7			88	8%	3	5.42	61.0	1.0	0.85	1.14	1.0	0.44 *	1.48
8	gi|115480295	6-phosphogluconolactonase, rice	262	20%	5	5.46	29.1	1.0	0.5 *	0.48 *	1.0	1.29	5.44 *
9	gi|46391135	ATP synthase beta chain, rice	277	17%	5	5.66	43.1	1.0	0.33 *	0.56	1.0	0.79	0.69
10	gi|56784991	ATP synthase beta subunit, rice	241	19%	6	5.33	45.9	1.0	0.87	0.38 *	1.0	0.98	0.75
11	gi|6815115	ATP synthase beta subunit, rice	700	21%	8	5.38	54.0	1.0	0.90	1.12	1.0	0.25 *	0.43 *
12	gi|11466784	ATP synthase CF1 alpha subunit, rice	85	4%	2	5.95	55.7	1.0	0.46 *	0.70	1.0	0.63	0.45 *
13	gi|11466784	ATP synthase CF1 alpha subunit, rice	655	23%	10	5.95	55.7	1.0	1.25	1.43	1.0	2.08 *	1.05
14	gi|11466784	ATP synthase CF1 alpha subunit, rice	512	21%	8	5.95	55.7	1.0	0.16 *	0.91	1.0	2.74 *	3.82 *
15	gi|115476908	ATP synthase D chain, mitochondrial, rice	77	21%	3	5.19	19.7	1.0	0.40 *	0.61	1.0	0.80	1.06
16	gi|194033257	ATP synthase F0 subunit 1, rice	330	16%	6	5.85	55.7	1.0	0.88	1.84	1.0	1.93	2.38 *
17	gi|110289207	Chaperonin CPN60-1, rice	158	5%	4	6.95	67.6	1.0	0.85	1.19	1.0	1.30	3.72 *
18	gi|115465974	6-phosphogluconate dehydrogenase, rice	376	13%	6	5.85	52.9	1.0	0.86	1.74	1.0	0.41 *	1.25
19	gi|2267006	Endosperm lumenal binding protein, rice	102	3%	2	5.30	73.7	1.0	1.01	1.23	1.0	0.37 *	0.22 *
20	gi|780372	Enolase, rice	321	12%	7	5.42	48.3	1.0	0.00 *	0.65	1.0	0.72	0.76
21	gi|115468758	Enoyl-CoA hydratase/isomerase, rice	152	8%	5	6.48	47.3	1.0	1.07	4.21 *	1.0	0.95	1.08
22	gi|115452127	Fructose-1,6-bisphosphatase, rice	216	8%	4	5.00	44.1	1.0	0.55	0.98	1.0	2.77 *	2.96 *
23	gi|115448167	Inorganic pyrophosphatase-like protein, rice	202	23%	4	5.56	24.4	1.0	0.98	1.12	1.0	0.48 *	0.67
24	gi|34393836	Ribose-5-phosphate isomerase, rice	249	22%	4	4.91	27.1	1.0	0.99	1.14	1.0	1.37	2.17 *
25	gi|27261025	Thiamine biosynthesis protein, rice	284	20%	7	5.44	37.2	1.0	1.13	0.01 *	1.0	1.32	1.68
26	gi|553107	Triosephosphate isomerase, rice	231	15%	4	6.60	27.8	1.0	1.07	1.44	1.0	1.40	2.48 *
27	gi|115469362	Vacuolar H+-ATPase subunit A, rice	410	9%	6	5.20	68.7	1.0	0.50	1.39	1.0	2.81 *	2.30 *
28	gi|62733870	Chlorophyll a/b-binding protein CP26, rice	243	28%	5	5.95	24.3	1.0	0.42 *	0.85	1.0	1.12	1.03
29	gi|255571784	Glutamate-1-semialdehyde 2,1-aminomutase, rice	77	8%	3	5.99	50.8	1.0	0.62	0.93	1.0	2.51 *	3.69 *
30	gi|32129323	Magnesium chelatase subunit chlD, rice	170	8%	6	5.38	81.0	1.0	0.68	0.99	1.0	2.05 *	1.23
31	gi|115448091	Phosphoribulokinase, chloroplast, rice	214	8%	4	5.68	45.2	1.0	0.56	0.71	1.0	0.37 *	0.40 *
32	gi|21839	Phosphoribulokinase, rice	45	2%	2	5.84	45.4	1.0	3.73 *	6.29 *	1.0	2.15 *	6.65 *
33	gi|34394725	Photosystem I reaction center subunit IV, rice	57	22%	2	9.64	15.5	1.0	0.29 *	0.30 *	1.0	0.99	2.07 *
34	gi|11466807	Photosystem II protein V, rice	93	26%	2	4.64	9.4	1.0	0.24 *	0.44 *	1.0	1.10	1.45
35	gi|11955	Ribulose-1,5-bisphosphate carboxylase, rice	103	6%	3	6.13	53.4	1.0	0.23 *	0.77	1.0	0.43 *	1.43
36	gi|11466795	Ribulose-1,5-bisphosphate carboxylase, rice	122	10%	4	6.22	53.4	1.0	0.65	2.82 *	1.0	0.35	1.15
37	gi|115439241	Similar to ATP synthase beta chain, rice	376	17%	7	6.10	59.6	1.0	0.45 *	1.05	1.0	0.79	1.19
38	gi|218210	Ribulose-1,5-bisphosphate carboxylase, rice	156	20%	3	9.11	19.8	1.0	0.29 *	1.59	1.0	6.43 *	3.78 *
39	gi|115456265	NAD-dependent epimerase/dehydratase, rice	124	11%	2	6.34	27.9	1.0	1.33	2.18 *	1.0	1.39	1.11
40	gi|4760553	Nad-dependent formate dehydrogenase, rice	207	14%	4	6.87	41.4	1.0	0.97	1.01	1.0	2.43 *	3.23 *
41	gi|115443619	NADH-hydroxypyruvate reductase, rice	237	14%	6	6.56	42.3	1.0	2.03 *	5.27 *	1.0	1.14	0.20 *
**Signal transduction**													
42	gi|115458806	14-3-3-like protein GF14-6, rice	376	27%	7	4.76	29.9	1.0	0.69	0.42 *	1.0	0.83	1.64
43	gi|108712139	Ankyrin repeat domain protein 2, rice	291	20%	5	4.64	37.3	1.0	0.86	1.68	1.0	1.22	2.21 *
44	gi|303859	Brain specific protein, rice	77	11%	2	4.77	29.2	1.0	0.69	1.05	1.0	0.81	7.12 *
45	gi|303859	Brain specific protein, rice	438	32%	6	4.77	29.2	1.0	0.48 *	0.35 *	1.0	2.01 *	2.63 *
46	gi|303859	Brain specific protein, rice	204	17%	5	4.77	29.2	1.0	0.35 *	0.40 *	1.0	1.33	3.09 *
47	gi|34393219	Calreticulin precursor, rice	221	7%	4	4.49	49.0	1.0	0.82	2.76 *	1.0	1.48	2.04 *
48	gi|115445587	GTP-binding protein typA, rice	186	7%	4	7.08	74.0	1.0	0.92	2.65 *	1.0	0.68	0.76
49	gi|12957710	GTP-binding protein, rice	460	18%	8	7.03	46.9	1.0	1.14	2.66 *	1.0	0.90	1.17
50	gi|6682927	Importin alpha 1b, rice	194	10%	4	5.18	59.1	1.0	0.67	1.17	1.0	1.02	4.29 *
51	gi|115471679	Spermidine synthase, rice	90	8%	2	5.23	35.6	1.0	0.75	2.01 *	1.0	1.00	1.39
52	gi|115476928	TaWIN2, rice	290	21%	4	4.85	29.1	1.0	0.89	1.92	1.0	3.00 *	1.29
53	gi|115456491	TolB, C-terminal domain protein, rice	191	5%	3	5.45	73.9	1.0	2.60 *	1.93	1.0	1.32	2.60 *
54	gi|115452585	Glutamyl endopeptidase, rice	162	5%	4	5.66	104.4	1.0	0.99	0.28 *	1.0	2.32 *	3.55 *
55	gi|115456491	TolB, C-terminal domain protein, rice	121	7%	4	5.45	73.9	1.0	1.27	4.11 *	1.0	1.10	1.47
56	gi|115458498	UVB-resistance protein UVR8, rice	79	5%	2	5.37	48.7	1.0	0.97	1.82	1.0	0.87	0.50 *
**Redox homeostasis**													
57	gi|115479659	Glutathione S-transferase GST 23, rice	111	15%	4	5.50	25.3	1.0	0.24 *	0.80	1.0	0.23 *	1.30
58	gi|11177845	Glutathione S-transferase OsGSTF3, rice	228	15%	4	5.81	25.1	1.0	0.86	0.65	1.0	1.37	2.21 *
59	gi|115477793	Glutathione S-transferase, rice	101	9%	3	6.84	37.5	1.0	0.32 *	0.45 *	1.0	0.77	0.76
60	gi|601869	Manganese superoxide dismutase, rice	469	35%	7	6.50	24.9	1.0	0.95	0.82	1.0	3.28 *	2.96 *
61	gi|50252391	Putative glyoxalase I, rice	103	8%	3	5.82	32.4	1.0	0.37 *	1.29	1.0	1.02	1.39
62	gi|538430	superoxide dismutase, rice	58	19%	2	5.71	15.3	1.0	0.42 *	0.73	1.0	2.41 *	1.94
63	gi|115475151	Lactoylglutathione lyase, rice	266	20%	5	5.51	32.9	1.0	0.38 *	0.63	1.0	0.59	1.05
64	gi|115439131	Peroxiredoxin, rice	87	16%	2	5.58	17.3	1.0	0.77	0.48 *	1.0	1.44	1.75
65	gi|115476190	Quinone oxidoreductase-like protein, rice	163	8%	3	7.63	39.7	1.0	0.43 *	1.04	1.0	0.36 *	0.38 *
66	gi|115111257	Betaine aldehyde dehydrogenase, rice	147	7%	3	5.29	55.4	1.0	0.75	0.95	1.0	2.37 *	2.06 *
67	gi|53370754	C1-like domain containing protein, rice	241	7%	5	5.37	85.6	1.0	0.71	0.81	1.0	0.94	5.31 *
68	gi|115456828	2-hydroxyacid dehydrogenase, rice	84	7%	2	6.00	34.7	1.0	0.96	4.20 *	1.0	0.30 *	0.47 *
69	gi|41052915	Ferredoxin-NADP(H) oxidoreductase, rice	222	19%	6	7.98	41.1	1.0	0.69	0.88	1.0	1.19	0.10 *
70	gi|41052915	Ferredoxin-NADP(H) oxidoreductase, rice	206	21%	6	7.98	41.1	1.0	0.34 *	0.92	1.0	1.98	1.13
71	gi|115461843	Isovaleryl-CoA dehydrogenase, rice	152	10%	5	6.52	45.1	1.0	0.46 *	0.48 *	1.0	1.82	1.89
72	gi|115474285	L-ascorbate peroxidase, rice	500	35%	6	5.21	27.2	1.0	0.50 *	0.61	1.0	0.84	0.62
73	gi|37718877	Methylenetetrahydrofolate reductase, rice	201	14%	4	6.10	42.2	1.0	0.37 *	0.74	1.0	0.95	1.25
74	gi|115453457	PDI-like protein, rice	270	12%	5	4.95	64.0	1.0	0.13 *	0.20 *	1.0	5.09 *	13.92 *
75	gi|115472727	Rieske iron-sulfur protein, rice	180	11%	3	8.54	24.2	1.0	0.53	0.47 *	1.0	1.50	1.26
76	gi|115436382	Thioredoxin domain 2 containing protein, rice	90	7%	2	6.43	41.1	1.0	1.23	0.84	1.0	2.45 *	1.52
77	gi|297728925	Thioredoxin, rice	294	23%	4	8.16	18.9	1.0	0.95	0.43 *	1.0	1.35	0.96
**Amino acid and nitrogen metabolism**													
78	gi|115455323	Cysteine synthase, rice	155	12%	3	5.35	34.4	1.0	0.42 *	0.92	1.0	1.24	1.36
79	gi|108862998	Cysteine synthase, rice	59	4%	1	8.76	43.8	1.0	1.08	0.97	1.0	0.92	3.38 *
80	gi|115442595	Cysteine synthase, rice	531	22%	7	6.28	42.1	1.0	0.30 *	0.51	1.0	1.23	1.09
81	gi|22748337	Glutamate ammonia ligase, rice	139	6%	2	5.73	38.8	1.0	0.41 *	0.95	1.0	4.37 *	1.36
82	gi|115465569	Ketol-acid reductoisomerase, chloroplast, rice	225	9%	4	6.01	62.7	1.0	0.26 *	1.48	1.0	1.89	1.47
83	gi|115452263	Ornithine acetyltransferase, rice	118	6%	2	6.45	48.3	1.0	0.71	0.12 *	1.0	1.38	1.10
**Carbohydrate metabolism**													
84	gi|218155	Chloroplastic aldolase, rice	267	11%	5	7.60	42.4	1.0	0.46 *	0.64	1.0	1.06	1.40
85	gi|218155	Chloroplastic aldolase, rice	456	14%	6	7.60	42.4	1.0	0.09 *	0.62	1.0	1.57	1.39
86	gi|115482534	Cytoplasmic malate dehydrogenase, rice	171	18%	4	5.75	35.9	1.0	0.26 *	0.37 *	1.0	1.66	1.97
87	gi|2497857	Malate dehydrogenase, mitochondrial, rice	320	12%	4	8.81	35.9	1.0	1.34	0.98	1.0	4.66 *	4.70 *
88	gi|115477843	Malate dehydrogenase, NADP-dependent, rice	188	6%	2	6.96	47.5	1.0	0.27 *	1.00	1.0	0.36 *	0.76
89	gi|4105561	Ribulose-5-phosphate-3-epimerase, rice	173	11%	2	8.93	29.2	1.0	0.75	2.30 *	1.0	0.48 *	0.77
90	gi|3024122	S-adenosylmethionine synthase 2, rice	378	18%	5	5.68	43.3	1.0	0.41 *	1.02	1.0	1.59	0.77
**Protein folding and assembly**													
91	gi|115479353	20 kDa chaperonin, chloroplast, rice	173	36%	5	5.97	25.5	1.0	0.81	0.96	1.0	1.24	0.23 *
92	gi|222631026	Chloroplast, Hsp70, rice	336	7%	4	5.12	73.7	1.0	1.00	1.59	1.0	1.95	0.30 *
93	gi|218161	Elongation factor 1 beta, *rice*	329	23%	5	4.86	23.8	1.0	1.28	1.00	1.0	1.99	2.36 *
94	gi|115489714	Glycine-rich RNA-binding protein 1, rice	506	45%	6	6.32	16.1	1.0	0.64	0.49 *	1.0	0.75	0.61
95	gi|27476086	Hsp 70, mitochondrial precursor, rice	288	8%	6	5.46	70.7	1.0	0.36 *	0.58	1.0	1.14	1.35
96	gi|115456247	Heat shock 70 kDa protein, rice	523	14%	9	5.10	71.7	1.0	0.98	1.16	1.0	5.64 *	1.18
97	gi|115448989	Heat shock 70 kDa protein, rice	406	13%	8	5.49	73.1	1.0	0.97	1.44	1.0	3.07 *	4.74 *
98	gi|18855040	Heat shock protein 90, rice	439	15%	9	4.89	93.0	1.0	0.48 *	1.21	1.0	1.33	1.27
99	gi|39104468	Heat shock protein 90, rice	579	19%	10	4.98	80.4	1.0	0.75	0.94	1.0	1.83	0.45 *
100	gi|39104468	Heat shock protein 90, rice	434	12%	8	4.98	80.4	1.0	0.70	0.26 *	1.0	6.29 *	2.32 *
101	gi|108862740	Hsp 90 protein, rice	122	5%	3	5.02	79.7	1.0	0.41 *	0.37 *	1.0	1.48	1.40
102	gi|115459670	Lambda integrase-like, rice	333	26%	6	5.57	25.8	1.0	0.96	2.64 *	1.0	0.76	0.49 *
103	gi|77557101	Methyl-CpG binding domain protein, rice	156	17%	4	4.74	31.5	1.0	0.91	1.55	1.0	2.07 *	1.26
104	gi|115470141	PDX1-like protein 4, rice	185	12%	5	6.41	33.9	1.0	0.82	1.02	1.0	0.72	2.31 *
**Protein hydrolysis**													
105	gi|42408435	Aminopeptidase N, rice	296	11%	7	5.42	98.6	1.0	0.75	0.42 *	1.0	1.02	1.23
106	gi|42408435	Aminopeptidase N, rice	152	4%	3	5.42	98.6	1.0	0.47 *	1.98	1.0	1.16	1.86
107	gi|42408435	Aminopeptidase N, rice	65	4%	2	5.42	98.6	1.0	0.35 *	0.37 *	1.0	1.51	1.30
108	gi|215694277	Aminopeptidase, rice	96	4%	3	5.38	100.1	1.0	2.46 *	1.14	1.0	1.30	1.35
109	gi|29468084	Aspartate aminotransferase, rice	391	18%	6	5.90	46.0	1.0	0.08 *	0.86	1.0	1.59	2.07 *
110	gi|115434854	Similar to Ubiquitin-specific protease 14, rice	94	1%	1	5.08	89.2	1.0	0.62	1.05	1.0	2.20 *	1.77
111	gi|115449043	Subtilisin-like protease, rice	192	7%	4	5.95	81.4	1.0	0.19 *	0.47 *	1.0	2.45 *	1.21
112	gi|115482934	Glycine cleavage system H protein, rice	88	8%	1	4.92	17.5	1.0	1.30	2.39 *	1.0	0.69	0.79
113	gi|8671508	Beta 4 subunit of 20S proteasome, rice	160	16%	3	5.42	23.6	1.0	0.41 *	1.07	1.0	0.37 *	1.22
114	gi|115444057	Proteasome subunit alpha type 1, rice	215	16%	3	5.37	29.9	1.0	0.71	0.99	1.0	1.08	3.16 *
115	gi|115447473	Proteasome subunit alpha type 2, rice	196	15%	4	5.39	25.8	1.0	0.76	0.87	1.0	1.05	2.60 *
116	gi|115486269	Proteasome subunit alpha type 5, rice	250	31%	5	4.70	26.1	1.0	0.37 *	0.48 *	1.0	1.58	3.14 *
117	gi|14091862	Putative hydrolase, rice	233	9%	3	9.17	41.4	1.0	0.93	0.64	1.0	1.86	2.37 *
**Protein synthesis**													
118	gi|115469770	60S acidic ribosomal protein P3, rice	86	22%	2	4.40	11.9	1.0	0.69	0.37 *	1.0	0.78	2.14 *
119	gi|6525065	Translational elongation factor Tu, *rice*	84	4%	2	6.05	50.5	1.0	0.41 *	0.65	1.0	4.07 *	7.31 *
120	gi|88866516	UDP-glucose pyrophosphorylase, *rice*	103	4%	2	5.43	51.8	1.0	1.15	1.78	1.0	2.21	1.56
121	gi|115449577	29 kDa ribonucleoprotein, chloroplast, rice	235	12%	4	5.17	34.9	1.0	0.78	0.72	1.0	0.40 *	0.63
122	gi|108707824	30S ribosomal protein S1, chloroplast, rice	380	18%	8	4.70	43.6	1.0	1.08	1.15	1.0	1.39	2.33 *
123	gi|115456525	30S ribosomal protein S6, chloroplast, rice	82	12%	2	7.79	23.4	1.0	0.48 *	0.80	1.0	2.10 *	1.06
124	gi|115456525	30S ribosomal protein S6, chloroplast, rice	121	16%	3	7.79	23.4	1.0	1.87	2.76 *	1.0	2.11 *	2.11 *
125	gi|115463659	50S ribosomal protein L1, rice	268	14%	3	6.86	38.9	1.0	0.71	0.80	1.0	2.08 *	2.06 *
126	gi|77548531	60S acidic ribosomal protein P0, rice	127	9%	2	5.38	34.5	1.0	0.15 *	0.82	1.0	1.60	1.79
127	gi|115442127	NAC-alpha-like protein 3, rice	88	16%	2	4.39	22.1	1.0	0.51	0.92	1.0	0.48 *	1.06
128	gi|115480705	Nucleic acid-binding protein precursor, rice	254	21%	6	4.41	35.4	1.0	0.77	1.09	1.0	3.36 *	1.97
129	gi|45433309	Nucleosome assembly protein 1, rice	125	10%	3	4.34	42.7	1.0	0.25 *	0.45 *	1.0	0.75	1.13
130	gi|115477393	Peptidylprolyl isomerase, rice	223	9%	4	5.15	64.4	1.0	1.04	1.01	1.0	1.39	0.34 *
131	gi|115437444	RNA helicase, rice	78	7%	3	7.03	56.5	1.0	0.81	4.01 *	1.0	1.97	0.83
132	gi|215769434	RNA-binding protein 8A, rice	196	30%	5	5.18	22.1	1.0	1.02	0.34 *	1.0	2.54 *	1.47
133	gi|125541223	Zn-dependent peptidases, rice	140	4%	4	5.41	121.9	1.0	0.74	1.40	1.0	0.19 *	0.14 *
**Others and unknown**													
134	gi|115477851	CaLB domain containing protein, rice	150	29%	9	4.77	30.4	1.0	0.76	1.02	1.0	0.91	2.70 *
135	gi|115450773	Cell division control protein 48 homolog A, rice	314	7%	6	5.12	90.4	1.0	0.43 *	0.24 *	1.0	6.91 *	1.55
136	gi|295885	Actin, rice	140	13%	4	5.29	42.2	1.0	0.44 *	0.86	1.0	0.10 *	1.12
137	gi|115434036	Isoflavone reductase, rice	205	15%	4	5.69	33.5	1.0	0.45 *	0.86	1.0	0.67	0.78
138	gi|115457122	O-methyltransferase, *rice*	376	17%	6	5.33	40.9	1.0	0.23 *	0.77	1.0	0.66	0.32 *
139	gi|4158221	Reversibly glycosylated polypeptide, rice	123	6%	2	5.82	41.9	1.0	0.16 *	0.67	1.0	1.67	0.97

^a^ Numbering of proteins expressed differentially in two rice cultivars. ^b^ Accession number from the database. ^c^ Names and species of the proteins obtained via the MASCOT software from the NCBInr database. ^d^ MOWSE score probability for the entire protein. ^e^ The sequence coverage of identified proteins. ^f^ The total number of identified peptide. ^g^ EpI is experimental isoelectric point. ^h^ Emw is experimental molecular mass. ^i^ The protein abundance ratio (Treatment/Control) at each particular time point. * Indicates significant (more than 2.0-fold or less than 0.5-fold) difference between control and treatment at 0.05 level.

## Data Availability

Not applicable.

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
