# Peer review of "Proteomic Analysis Reveals Salicylic Acid as a Pivotal Signal Molecule in Rice Response to Blast Disease Infection"

_plants, 2022, doi:10.3390/plants11131702_

Round 1

Reviewer 1 Report

Manuscript submitted by Zhou et al. entitled “Proteomic analysis reveals salicylic acid as a pivotal signal molecule in rice response to blast disease infection” aims to provide a blast fungus responsive protein associated with various molecular mechanisms, especially SA mediated regulation, using two different varieties of rice. However, the author needs to address many issues with the manuscript to make it acceptable for publication.

- English needs to be revised.

- Some key explanation and citation regarding experiments are missing or not well described. In case of proteome analysis part, author must be clearly mention whether this proteome analysis performed by 2-DE or label-free or phosphoproteome approaches. Please carefully described which kind of approach was performed with iTRAQ labeling analysis. Moreover, letters in most of figures are not well visible. Figure preparation should be improved.

- Line. 99: Author need to provide appropriated abbreviations and explanation of each sample.

- Line. 115: kDa polypeptide of photosystem II -> 23kDa polypeptide of photosystem II

- Line. 129: R- -> R-SA in Table 1

- Table 1: The intensity values of R and M samples are relative intensity? Then please mention it with separate column.

- Table 1: Table contains a lot of duplicate of identified proteins with different sequence coverage and score values. And these proteins showed mixed regulation (up and down-regulated) as compared with CK sets. I wonder whether these proteins are actually differentially expressed or not as comparison among different sample sets.

- Line. 147 to 152 (Figure 3): This paragraph and Venn diagram are difficult to understand. In case of isobaric labeling approach for protein identification, they can apply quantification with labeled peptides only. So, this Venn diagram indicated the number of identified proteins in each sample OR the number of differentially expressed (increased/decreased) proteins in each sample? Please clearly mention it on section 2.2 and figures. Moreover, please clearly mention whether most of DEPs on R-M (24) sample are down-regulated after infection of blast fungus, therefore, it assumed negative effect due to blast infection or not.

- Line.157: Phosphoproteome and 2-DE analysis also performed? Please check and revise it.

- Line. 177: Section 2.4 must be carefully re-arranging for better understand of data. Whole this section still contains some point of errors. For instance, in line 184, author wrote “significant differences were observed when comparing two varieties". However, it is confusing that which kind of comparisons are performed. Moreover, in line 188 to 189, which kind of proteins are highly up-regulated? Please carefully check and mention it.

- Line. 185: Please remove the sentence “in the R-M and R-SA groups”

- Line. 241: Proteomics -> Proteomic

- In discussion part, author wrote “proteins spots” in all the sections. However, this study performed iTRAQ labeling analysis for quantification and identification of differentially expressed proteins. Please remove “spot” or replace another appropriated word.

- Line. 450: Please provide appropriated citation regarding normalization approach for iTRAQ labeling proteomics.

- Line. 459: a total of “phosphoproteins” or “proteins”. Please check and revise it.

- Please clearly mention narrow-down approaches and provide all statistical data in Table or Supplementary table.

- All proteome raw files must be deposit to PRIDE or other available website.

Reviewer 2 Report

Rice blast caused by fungus Magnaporthe grisea is one of the most destructive diseases in rice production worldwide, and salicylic acid can efficiently decrease the damage of rice blast. Authors results suggest that the glycolytic pathway is attested in blast-susceptible Nip-ponbare under blast infection, whereas the blast infection resistance of Minghui is correlated with salicylic acid-mediated regulation. Overall, the manuscript is well written, therefore it can be accepted for publication the present form after a careful English language check.

Reviewer 3 Report

              In the work entitled 'Proteomic analysis reveals salicylic acid as a pivotal signal molecule in rice response to blast disease infection' the posttranslational responses of rice blast-resistant Minghui and blast-susceptible Nipponbare cultivars under blast fungus infection using the iTRAQ approach was examined, intending to explain the molecular mechanisms underlying blast resistance in rice. I don't like Abstract. It should contain more specific information. In this form, it is too general. The Introduction chapter presents all the necessary information. In the Materials and Methods chapter, the part concerning the analyzed material is not clearly written. It is not clear how the experiment was designed and performed. The description should be supplemented with a table, and all used abbreviations should be explained. For example, it is not known whether SA spraying preceded fungus infestation or vice versa. The rest of this chapter is correct. There are only a few errors that I have marked in the attached file. The Results and Discussion sections were also properly edited. Errors appear in the citation of the publication, in the spelling of Magnaporthe grisea, which should be written in italics. The nomenclature of individual variants introduces a bit of confusion. I mean the different varieties and different ways of treating them with Magnaporthe grisea or salicylic acid. I don't like the Conclusions chapter. First of all, the text is bad to read. The sentences have a strange structure. This chapter should be edited. This is very often read by readers and needs to be improved. The manuscript can be published after minor revision.
